# Broad Cross-Reactive IgA and IgG against Human Coronaviruses in Milk Induced by COVID-19 Vaccination and Infection

**DOI:** 10.3390/vaccines10060980

**Published:** 2022-06-20

**Authors:** Jiong Wang, Bridget E. Young, Dongmei Li, Antti Seppo, Qian Zhou, Alexander Wiltse, Anna Nowak-Wegrzyn, Katherine Murphy, Kaili Widrick, Nicole Diaz, Joseline Cruz-Vasquez, Kirsi M. Järvinen, Martin S. Zand

**Affiliations:** 1Department of Medicine, Division of Nephrology, University of Rochester, Rochester, NY 14642, USA; jiong_wang@urmc.rochester.edu (J.W.); qian_zhou@urmc.rochester.edu (Q.Z.); awiltse@som.umaryland.edu (A.W.); 2Department of Pediatrics, Division of Allergy and Immunology, University of Rochester, Rochester, NY 14642, USA; bridget_young@urmc.rochester.edu (B.E.Y.); antti_seppo@urmc.rochester.edu (A.S.); katherine_murphy@urmc.rochester.edu (K.M.); kaili_widrick@urmc.rochester.edu (K.W.); nicole_diaz@urmc.rochester.edu (N.D.); 3Clinical and Translational Science Institute, University of Rochester, Rochester, NY 14642, USA; dongmei_li@urmc.rochester.edu; 4Department of Pediatrics, Division of Pediatric Allergy and Immunology, Hassenfeld Children’s Hospital, NYU Grossman School of Medicine, New York, NY 10016, USA; anna.nowak-wegrzyn@nyulangone.org (A.N.-W.); joseline.cruzvasquez@nyulangone.org (J.C.-V.); 5Department of Pediatrics, Gastroenterology and Nutrition, Collegium Medicum, University of Warmia and Mazury, 10-719 Olsztyn, Poland

**Keywords:** human milk IgA and IgG antibodies, SARS-CoV-2 antibody response, human common coronaviruses (HCoVs), cross-reactive antibodies, COVID-19 vaccine, COVID-19 infection

## Abstract

It is currently unclear if SARS-CoV-2 infection or mRNA vaccination can also induce IgG and IgA against common human coronaviruses (HCoVs) in lactating parents. Here we prospectively analyzed human milk (HM) and blood samples from lactating parents to measure the temporal patterns of anti-SARS-CoV-2 specific and anti-HCoV cross-reactive IgA and IgG responses. Two cohorts were analyzed: a vaccination cohort (n = 30) who received mRNA-based vaccines for COVID-19 (mRNA-1273 or BNT162b2), and an infection cohort (n = 45) with COVID-19 disease. Longitudinal HM and fingerstick blood samples were collected pre- and post-vaccination or, for infected subjects, at 5 time-points 14–28 days after confirmed diagnosis. The anti-spike(S) and anti-nucleocapsid(N) IgA and IgG antibody levels against SARS-CoV-2 and HCoVs were measured by multiplex immunoassay (mPlex-CoV). We found that vaccination significantly increased the anti-S IgA and IgG levels in HM. In contrast, while IgG levels increased after a second vaccine dose, blood and HM IgA started to decrease. Moreover, HM and blood anti-S IgG levels were significantly correlated, but anti-S IgA levels were not. SARS2 acute infection elicited anti-S IgG and IgA that showed much higher correlations between HM and blood compared to vaccination. Vaccination and infection were able to significantly increase the broadly cross-reactive IgG recognizing HCoVs in HM and blood than the IgA antibodies in HM and blood. In addition, the broader cross-reactivity of IgG in HM versus blood indicates that COVID-19 vaccination and infection might provide passive immunity through HM for the breastfed infants not only against SARS-CoV-2 but also against common cold coronaviruses.

## 1. Introduction

The severe acute respiratory syndrome virus 2 (SARS-CoV-2) is responsible for coronavirus disease 2019 (COVID-19) pandemic, causing more than 5 million deaths worldwide as of January 2022 [1]. SARS-CoV-2 belongs to the zoonotic *coronaviridae* family [2]. The human coronaviruses include SARS-CoV-1 (SARS-CoV), Middle Eastern Respiratory Syndrome (MERS-CoV), and the endemic β- (HCoV-OC43, HCoV-HKU1) and α-human cold coronaviruses (HCoV-229E, HCoV-NL63) [3], which are responsible for 30% of mild upper respiratory infections [4,5]. The major immunodominant epitopes for HCoVs are located on the homotrimeric spike (S) glycoprotein [2], and SARS-CoV-2 S shares these sequence homologies with other β-HCoVs. The S-protein N-terminal S1 subunit has a receptor-binding domain (RBD), mediating viral binding via high-affinity interactions with host cell angiotensin-converting enzyme 2 (ACE2). In contrast, the S2 subunit is responsible for virus-cell membrane fusion [6], and displays more sequence homology between HCoV strains than the S1 subunit [7,8].

Individuals with symptomatic COVID-19 infection exhibit increases in an extensive range of anti-SARS-CoV-2 antibodies [6,9,10]. The patterns of seroconversion in most individuals are similar to those of secondary immune responses, and rapid and robust antibody response was correlated with disease severity [6]. Our previous clinical study suggested that pre-existing OC43-reactive antibodies are involved in the early humoral response to SARS-CoV-2 [11]. Other groups have reported that pre-existing cross-reactive antibodies or memory B cell immunity might be protective against COVID-19 [12,13,14].

Human milk (HM) provides protection against various infectious diseases, including respiratory illnesses in infants [15]. This protection is in part due to the passive immunity transferred via maternal immunoglobulins [16]. Anti-SARS-CoV-2 secretory IgA is present in HM during and after acute infection [17,18], and those antibodies neutralize SARS-CoV-2 in vitro [19,20]. Currently, COVID-19 mRNA vaccination is the most effective way to prevent SARS-CoV-2 infection and transmission [21]. Studies have also shown that mRNA vaccination can elicit high titers of IgA and IgG antibodies in HM [22,23], which have neutralizing activity against SARS-CoV-2 [20]. The majority of evidence suggests that transmission of SARS-CoV-2 is overwhelmingly airborne, with minimal evidence of transmission via HM [24,25]. Thus, passive transfer of maternal antibodies in HM may provide neonatal protection from SARS-CoV-2 infection and mitigate severe disease.

Antibody cross-reactivity across different viral strains is an important source of pre-existing immunity to emerging viral variants [10,26,27]. Similarly, vaccination against a new viral strain may “back-boost” cross-reactive immunity to previously circulating strains [28,29]. This appears to be the case for SARS-CoV-2. It has been reported that pre-pandemic HM samples exhibited low-level cross-reactivity to the RBD subunit of SARS-CoV-2 [18]. However, it is unclear whether SARS-CoV-2 infection or vaccination can produce antibodies against common HCoVs in HM or blood, and the characteristics of any such elicited cross-reactive antibodies. This study quantitatively assessed anti-S and anti-N HM IgA and IgG antibodies against SARS-CoV-2, SARS-CoV-1, and four other common HCoVs (OC43, HKU1, NL63, 229E) following COVID-19 infection and vaccination.

## 2. Materials and Methods

### 2.1. Study Cohorts

This study was approved by the Institutional Review Boards (IRB) at the NYU Grossman School of Medicine (IRB i20-00601) and University of Rochester Medical Center (IRB STUDY00004889). All participants provided informed consent. All subjects’ information and research data were coded in compliance with the Department of Health and Human Services Regulations for the Protection of Human Subjects (45 CFR 46.101(b) (4)).

*Infected cohort*. Forty-six lactating parents with COVID-19 infection were recruited nationally between July 2020 - April 2021. To be eligible, participants needed to have an RT-PCR COVID-19 diagnosis within the previous 14 days and be providing HM to an infant ≤ 6 months of age. The eligible subjects aged ≥ 18 years collected milk samples at home on days 0, 3, 7, 10, and 28 as described below.

*Vaccinated cohort*. Thirty participants scheduled to receive their first dose of either Pfizer-BioNTech/BNT162b2 or Moderna/mRNA-1273 between December 2020–January 2021, and lactating with an infant of any age. Known, previous COVID-19 diagnosis of mother or infant was an exclusion criterion. milk and fingerstick samples were collected prior to receiving the first dose of vaccine (Pre), 18 days after the first dose (Vac1), and 18 days after the second dose (Vac2).

### 2.2. Sample Collection

See Figure 1 for details. *HM samples:* All participants collected milk and fingerstick capillary blood samples at home as previously described [20]. 5–10 mL HM samples were stored in the home −20 °C freezer until transport to the lab, packaged on ice via overnight mail. Milk samples were spun at 10,000× *g* for 10 min at 4 degrees to remove the skim milk in the lab, and were aliquoted and stored at −80 °C until analysis.

*Fingerstick blood samples:* All capillary blood samples were collected with volumetric absorptive microsampling (VAMS) 10 μL device (Mitra Collection Kit; Neoteryx, CA, USA) as previously published [30]. Each subject collected two 10 μL separate volumetric swabs, for a total of 20 μL of blood. All VAMS tips were placed in sealed containers with silica desiccant packets immediately after sampling, and stored in the home −20 °C freezer until returned to the lab with the HM samples. Once in the lab, samples were stored at −20 °C until analysis. To extract the antibodies, VAMS tips were individually soaked in 200 μL extraction buffer (PBS + 1% BSA + 0.5% Tween) in 1 mL deep 96 well plates (Masterblock, GBO, Austria) and shaken overnight. The eluant was stored at 4 °C and analyzed by multiplex assay within 24 h.

### 2.3. mPlex-Cov Assay

The mPlex-CoV assay, developed in our laboratory, is a Luminex-based immunoassay that can simultaneously estimate antibody concentrations against SARS-CoV-2 and other HCoVs from fingerstick blood samples. Briefly, in-house expressed and purified trimerized S protein and N protein of SARS-CoV-2, as well as recombinant S1, S2, and RBD domain proteins (SinoBio, Beijing, China), were coupled to magnetic microsphere beads (Luminex, Austin, TX, USA) at a concentration of 40 pmole/10^6^ beads using the xMAP® Antibody Coupling kit (Luminex, Austin, TX, USA). For the assay, 200 μL of eluent from VAMS devices (1:20) was further diluted 1:50 (IgG assay), or 1:10 (IgA) to yield final sample dilutions of 1:1000 (IgG) and 1:200 (IgA). A 50 μL volume of each diluted sample was added to duplicate wells of black, clear-bottom 96 well plates (Microplate, GBO, Kremsmünster, Austria), with 50 μL of the mPLeX-CoV bead panel added to each well, as previously described [11,30]. After washing, goat anti-human PE-conjugated IgG and IgA secondary antibodies (Southern Biotech, Birmingham, AL, USA, Cal No:2040-09, 2050-09) were used as the detection reagent. The calculation of IgG antibody concentrations against each HCoV virus strain was performed using the Bio-Plex Manager™ 6.2 software (Bio-Rad Co., Hercules, CA, USA) with standard curves of IgG and IgA generated by in-house HCoV standard positive serum samples (CoV-STD) [30].

### 2.4. Data Analysis and Statistical Methods

Spearman’s correlations were used to calculate the correlation matrix of the IgA and IgG antibodies concentrations against SARS-CoV-2 (SARS2) full spike, S1, S2, and RBD in both milk and blood. Spearman’s correlations of IgA and IgG antibody levels between milk and blood were also calculated. Similarly, the correlation matrix of milk and blood anti-S IgA, IgG concentrations against SARS2, SARS-CoV-1(SARS1), and other HCoVs elicited by vaccination and acute COVID-19 infection were also calculated using Spearman’s correlation. Spearman’s tests were used to examine whether the calculated Spearman’s correlations were significantly different from zero. The Spearman’s correlations and tests were conducted using statistical analysis software R version 4.0.2 (R Core Team, 2017).

The longitudinal anti-S, S1, S2, RBD, and anti-N SARS-CoV-2 specific IgA and IgG antibody responses elicited by COVID-19 mRNA vaccination in milk and fingerstick blood samples were examined using the generalized linear mixed-effects models. Similarly, the longitudinal milk and blood HCoV-reactive IgA and IgG antibody levels against spike proteins of SARS-CoV-2, SARS-CoV-1, OC43, HUK1, 229E, and NL63 were also estimated through generalized linear mixed-effects models. Pairwise comparisons within the generalized linear mixed-effects model framework were used to examine the differences in antibody levels over time. Statistical analysis software SAS V9.4 (SAS Institute Inc., Cary, NC, USA) was used for fitting the generalized linear mixed-effects models and corresponding pairwise comparisons.

The significance levels for all tests were set at 5% with two-sided tests.

## 3. Results

### 3.1. Study Cohorts and Samples

The research cohorts and plan were presented in Figure 1. The participant’s characteristics listed in the Appendix A have been previously reported [20]. This study was approved by the Institutional Review Boards (IRB) at the University of Rochester Medical Center and New York University Grossman School of Medicine. All participants provided informed consent.

### 3.2. Vaccination Elicited Strong SARS-CoV-2 Specific IgA and IgG Responses in Blood and Milk

We first estimated anti-SARS-CoV-2 IgA and IgG antibody responses against whole S protein (SARS-CoV-2-S), S1, S2, and RBD subunits elicited by mRNA vaccination. IgA and IgG in human milk (HM) and blood significantly increased by 18 days after the first dose of SARS-CoV-2 mRNA vaccine (Vac1) compared with pre-vaccination (*p* < 0.0001) (Figure 2), consistent with previous reports for HM [20,22,31,32] and our recent findings of blood IgG responses in healthy adults [11]. Except for one participant, who had higher milk anti-S and anti-N IgG levels before vaccination consistent with previous SARS-CoV-2 exposure, HM anti-S levels uniformly increased 10 (IgA) and 100 fold (IgG) post-vaccination. However, milk and blood anti-S IgA levels remained constant or decreased after the second vaccine dose (Vac2). In contrast, HM and blood IgG levels continued to increase compared with levels after Vac1 (*p* < 0.0001), which could persevere for up to 187 days after Vac2, as shown by our previous study [20]. In addition, the anti-S2 subunit-specific IgG levels did not increase as high as IgG against the full S protein, S1, and RBD sub-regions. As expected, we did not detect N-specific antibody responses following vaccination (Figure 2A).

To understand the relationship between the vaccine-induced SARS-CoV-2 reactive IgA and IgG responses in blood versus HM, we performed a correlation analysis (Figure 2B). Significant positive correlations exist between HM and blood IgA and IgG antibodies against full-spike protein, S1, S2, and RBD subunits (Pearson’s r in the range 0.22–0.46, Figure 2B). Elevated IgG antibody levels after Vac2 showed a higher correlation between milk and blood compared to Vac1, but r values decreased after Vac2. Overall, milk IgA levels did not correlate with blood IgG (*p* > 0.05). In contrast, milk IgG had the highest correlation with blood IgG (mean r = 0.36, *p* < 0.05). This suggests a common source for milk and serum IgG, but a different source for milk IgA, consistent with our previous work [33]. It is noteworthy that anti-SARS-CoV-2-S, S2 and RBD IgA concentrations in milk did not correlate with concentrations in blood after Vac1. This lack of correlation strongly suggests that IgA in milk originates from a pool of mucosal B-cells and is distinct from serum IgA.

### 3.3. COVID-19 Infection Significantly Increases SARS-CoV-2 Specific IgA and IgG Antibody Levels in Blood and Milk

Previous reports have indicated a wide inter-individual variability of anti-S SARS-CoV-2 IgG antibody response between participants with infection [20]. To concisely evaluate the milk IgA and IgG response to infection, we categorized the 45 subjects into three groups based on the ratio of milk anti-SARS-CoV-2 S IgG levels between day 0 (D0) vs. day 28 (D28). The *Increasing* group (n = 23) had D28/D0 antibody concentration ratios > 1.2. The *Constant* group (n = 10) had a ratio of D28/D0 antibodies between 0.8–1.2. Finally, the *Decreasing* group (n = 12) had a ratio of D28/D0 antibodies < 0.8. Anti-S, anti-N of IgA and IgG antibodies in milk are shown in Figure 3A, as well as antibodies that recognized S1, S2, and RBD-subunits of spike proteins.

These results demonstrate that SARS-CoV-2 infection can significantly increase anti-S IgA and IgG levels in most subjects compared with the naive healthy pre-vaccination cohort; only one subject’s milk IgG remained undetectable over the 28 days. In the Constant and Decreasing groups, antibody levels of day 0 vs. day 25 do not significantly change (*p* > 0.05). Other studies have documented the appearance of anti-S IgG 7–10 days after infection [6,10]; IgM and IgA appear even earlier. Our subjects were enrolled 0–14 days after COVID-19 diagnosis (average: 8 days), and we were able to detect an upward trajectory of IgG antibody levels in half of the subjects (the Increasing group). Compared to anti-S IgA, IgG concentrations did not exhibit such robust changes from day 0–28. Given that we collected our first sample (Day0) at an average 8 days post diagnosis, we likely missed the initial surge in IgA antibodies, which has been shown to occur earlier than the increase in milk IgG [20]. This pattern is also consistent with the trajectory of HM IgA and IgG after vaccination and other reports [32,34,35].

To our knowledge, this is the first study to report milk anti-N IgA and IgG levels in infected lactating participants. Our results show that acute infection elicited significant anti-N IgA and IgG antibodies in HM, with very similar trajectories post-infection. However, the change in anti-N IgA was more uniform among enrolled subjects than that of anti-S IgA, with an average 26.7 and 13.1 fold increase (equals [Con${}_{t_{2}}$]/[Con${}_{t_{1}}$] − 1), respectively, across all time-points. In contrast, the HM anti-S IgG changes were much greater than the increase of anti-N IgG antibodies, and the average fold changes were 87.4 and 8.6, including all post-infection samples, respectively. The blood anti-S and -N IgA and IgG antibody responses to the infection are shown in Appendix A. While anti-S and -N IgG antibodies kept increasing from the enrollment to 28 days after, IgA levels remained steady.

We also performed a correlation analysis of IgA and IgG antibodies in HM and blood in the Infection cohort. Results indicated that the IgA and IgG antibodies were highly correlated between blood and milk (Figure 3B). During acute infection, SARS-CoV-2 specific IgG antibodies against S, S1, S2, and N, were highly correlated with IgG antibodies in the blood (r = 0.48–0.85; Figure 3B), suggesting that the milk SARS-CoV-2 reactive IgG most likely are blood IgG related. The HM IgA also showed a high-level correlation with concentrations in the blood (r = 0.33–0.76; *p* < 0.001). Furthermore, HM anti-SARS-CoV-2 IgA levels significantly correlated with blood IgG levels, which was not the case with vaccination. Milk IgG levels correlated with blood IgA levels, at lower r values. The correlations between HM and blood IgA (upper left quadrant of correlation matrix in Figure 3B), between HM IgA with blood IgG (lower left quadrant), and between HM IgG with blood IgA (upper right quadrant) decreased from 0 to 28 days (*p* < 0.05). However, the HM IgG levels were still highly correlated with blood IgG at day 28. These data suggest a coupled response between IgG and IgA between HM and blood in infection which includes a mucosal response seen also in HM which was not the case with vaccination. This is expected as intramuscular vaccination does not have mucosal involvement whereas COVID-19 disease does.

Overall, the milk anti-SARS-CoV-2 virus IgA and IgG antibody showed higher correlation r values during the acute infection (day 0) than 28 days later. Additionally, correlations between the milk and blood antibodies are higher in the acute infection cohort than in the vaccination cohort. Moreover, the above results suggest that the lactating participant can produce anti-SARS-CoV-2 IgA and IgG antibodies during acute infection, and that IgG antibodies persist at elevated concentrations out to at least 28 days. Of clinical note, our previous work establishes that these antibodies can neutralize live SARS-CoV-2 in-vitro, likely conferring clinical protection to the recipient infant [20].

### 3.4. Anti-HCoV Cross-Reactive IgA and IgG Antibodies in Milk Elicited by COVID-19 Vaccination and Infection

As described above, HM anti-SARS-CoV-2 IgG increased robustly after first dose and second dose of vaccination but IgA increased more modestly (Figure 4A,B). The HM anti-SARS-CoV-2 IgG robustly increased after the first dose (163 fold compared to pre-vaccination control) and second doses (780 fold) of mRNA vaccination, but anti-SARS-CoV-2 IgA only increased 5.21 and 2.11 fold, respectively. Interestingly, vaccination also resulted in significant elevation of IgG antibodies that recognize SARS1, OC43, HKU1, 229E, and NL63 viruses in HM, with a 0.43–6.04 fold increase. However, it was not the case in blood where the vaccination induced a broad-reactive IgG only against OC43 and HKU1 (0.12–0.45 fold increase), but did not induce IgG to 229E and NL63, two α-HCoV that are more antigenically divergent.

During infection, the similar cross-reactive IgA and IgG antibody binding patterns were observed in the HM and blood samples (Figure 4C,D). Infection caused the HM anti-SARS-CoV-2 IgA after day 0 and day 28 to increase 7.15 and 5.56 fold, and IgG 28.5 and, 71.00 fold, respectively. Notably, the HM and blood IgA and IgG showed broader and stronger cross-reactive binding to the S proteins of OC43 and HKU1 β- HCoVs than those induced by vaccination. Surprisingly, HM IgG induced by SARS-CoV-2 infection showed broader cross-reactivity than HM IgA and blood IgG, including reactivity against NL63 and 229E, two α-HCoVs.

Additional correlation analysis was performed between the antibodies in HM versus blood, and IgG versus IgA. HM IgG was highly correlated to blood IgG levels after vaccination, whereas infection caused a higher correlation between HM and blood and IgG with IgA, especially in the early phase of COVID-19 (Figure 5). Additionally, anti-β-HCoV antibodies are more correlated to each other than to anti-α-HCoV antibodies, and anti-SARS2 S2 subunit antibodies are highly correlated to all β-HCoV S antibodies.

## 4. Discussion

This is the first study to document that in human milk (HM) both COVID-19 mRNA vaccination and SARS-CoV-2 (SARS2) infection induced broadly cross-reactive IgG and IgA antibodies against human common cold coronaviruses (HCoVs), as estimated by multiplex immunoassay (mPlex-CoV) [30]. In general, we found that the acute infection elicited a broader cross-reactive antibody response against HCoV spike proteins compared to the two dose mRNA vaccinations. Furthermore, our results demonstrate a high correlation between broadly cross reactive anti-S IgG in HM and peripheral blood broad post-infection and vaccination, with no milk-blood correlation for anti-S IgA. Interestingly, HM cross-reactive IgG demonstrated broader and higher magnitude cross-reactivity to S proteins tested, including OC43 and HKU1 β-HCoVs, 229E and NL63 α-HCoVs. These result highlight the question of the origin of such broadly cross-reactive anti-coronavirus S protein IgG and IgA.

Mucosal IgA is produced by plasma cells in the lamina propria and are transported across epithelial cells by the polymeric immunoglobulin receptor (pIgR) [36]. Milk IgA is produced by mammary gland B cells that have migrated from the mother’s intestine via the “enteromammary link” [37], as shown in animal studies [38,39,40,41]. This is controlled by the mucosal vascular addressin, MadCAM-1, which interacts with the gut homing receptor α4β7 integrin [42] and mucosal-associated CCR10 [43]. Furthermore, in a rabbit model, either oral or inhaled RSV resulted in RSV-specific IgA production in milk, bronchial and enteral secretions, whereas systemic immunization did not [44]. Human studies have shown that oral immunization in women results in an increase in plasma cells specific to non-pathogenic E. coli strains in milk, but not in saliva or blood [45]. This compartmentalization of mucosal IgA secreting B cells may explain the lack of correlation between HM and blood IgA anti-HCoVs post-infection or vaccination.

In contrast to IgA, we found a high correlation between the anti-S HCoV binding profiles in IgG. This may be due to the dichotomous source of HM IgG: In HIV, for example, it has been demonstrated that specific IgG secreting cells can be found in HM, which predominate over IgA-secreting cells and have a mucosal homing profile similar to gut-associated lymphoid tissue B cells [46]. Moreover, specific anti-HIV IgGs isolated from HM and plasma have similar neutralization potencies, despite lower HM concentrations, suggesting that the neutralization responses in HM are mainly due to plasma-derived IgG [47]. Therefore, Neutralizing capacity of HM IgG may be a function of the ratio between specific anti-HIV IgG produced by mucosal versus peripheral blood IgG secreting cells, which may vary between mothers [48].

The source for the broadly cross-reactive anti-S IgG antibodies found in HM is not precisely known. HM IgG antibodies are produced by B cells, which originate in the gut- or bronchus associated lymphoid tissue and home to HM, or circulate in plasma during respiratory infections [47]. Broad cross-reactivity with seasonal human coronaviruses may indicate that that this IgG production is boosted in the airways via mucosal associated memory B cells within lymphoid tissue at the site of previous exposure to seasonal coronaviruses. This would be a novel mechanism providing broadly crossreactive antibodies. Irrespective of their origin, these anti-SARS-CoV-2 IgA and IgG antibodies potentially serve to provide broadly protective antibodies to the infant.

It is well known that the S2 subdomain of spike protein shares more homology with other HCoVs than the S1 subdomain, including β- (SARS-CoV-1, OC43, and HKU1), and α- (229E and NL63) HCoVs [2,3,5]. Studies have also shown that SARS-CoV-2 infection can enhance preexisting HCoV immunity through anti-S2 IgG and memory B cell formation [14]. During infection, a rapid increase in S2-reactive IgG levels highly correlates with a rapid increase SARS-CoV-2 IgG and, counter-intuitively, the severity of COVID-19 disease [11]. Thus, it remains unclear, whether a predominance of anti-S2 IgG in milk contributes substantially to passive infant protective immunity to SARS-CoV-2 exposure in infants. In this study, we found a high correlation between the IgG antibodies against SARS-CoV-2 with β-HCoVs IgG in milk and blood (Figure 5), after vaccination and after infection. Interestingly, the high correlation between SARS-CoV-2 S2-reactive and β-HCoV-reactive IgA occurred only in the post-infection cohort, and not after vaccination. This suggests that infection may elicit a broader cross-reactive IgA response in milk than vaccination. Further study is needed to determine if anti-S2 predominant SARS-CoV-2 IgA in milk provides neutralizing protection against exposure.

A strength of this study is the longitudinal collection of human milk (HM) and fingerstick blood samples. Especially for the vaccination cohort, each subject had the pre- and post- vaccination samples at multiple time points. The mPlex-Cov assay used in this study to estimate the binding activities of cross-reactive antibodies in the milk and blood samples after vaccination and infection. The limitation of low antibody concentration of the fingerstick samples would be a technical challenge to measure the neutralizing antibodies against the OC43, HKU1 in in vitro assays. However, our previous work [20] demonstrated that milk neutralizing anti-SARS-CoV-2 antibodies could be induced by COVID-19 mRNA vaccination and SARS-CoV-2 infection.

## Figures and Tables

**Figure 1 vaccines-10-00980-f001:**
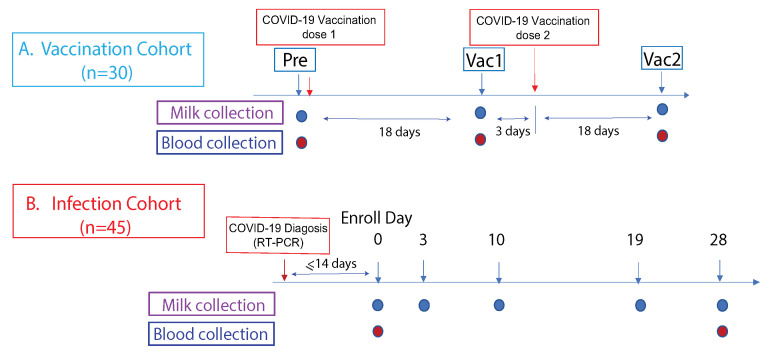
Study cohorts and experimental design. This prospective study consist of two cohorts: (**A**) The vaccination cohort enrolled lactating parents (n = 30) older than 18 years without COVID-19 infection history and scheduled to receive either the Pfizer-BioNTech/BNT162b2 or Moderna/mRNA-1273 mRNA vaccination. The human milk and fingerstick blood samples were collected before vaccination and 18 days after the first and second doses each. (**B**) The infection cohort enrolled lactating parents who had received an RT-PCR COVID-19 diagnosis within the previous 14 days. The human milk samples were collected at enrollment day 0, then on days 3, 10, 19, and 28. Fingerstick blood samples were collected on days 0 and 28.

**Figure 2 vaccines-10-00980-f002:**
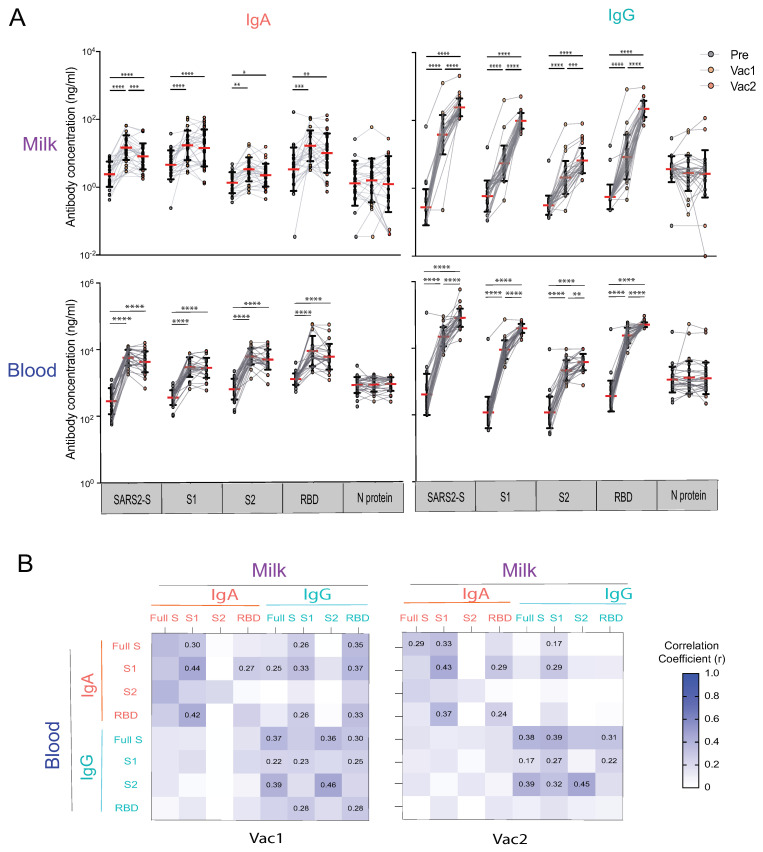
Human milk and blood SARS-CoV-2 specific IgA and IgG antibody response to mRNA vaccination in lactating parents. (**A**) IgA and IgG antibody responses to SARS-CoV-2 S (SARS2-S), S1, S2, RBD, and N of SARS-CoV-2 elicited by COVID-19 mRNA vaccination. The milk and fingerstick blood samples were collected pre-vaccine (PRE), 18 days after the first dose (Vac1), and 18 days after the second dose (Vac2). Antibody concentrations were estimated using the mPLEX-CoV assay (see Methods). Generalized linear mixed-effects models were used to test for differences between time points (**** *p* < 0.0001, *** *p* < 0.001, ** *p* < 0.01, * *p* < 0.05). (**B**) Heatmap of the Spearman correlations between IgA and IgG concentrations against SARS2-S, S1, S2, and RBD in milk and blood. Correlation coefficients (r) are color coded as shown in the figure and numerical values are given when the correlation p-value are less than 0.005.

**Figure 3 vaccines-10-00980-f003:**
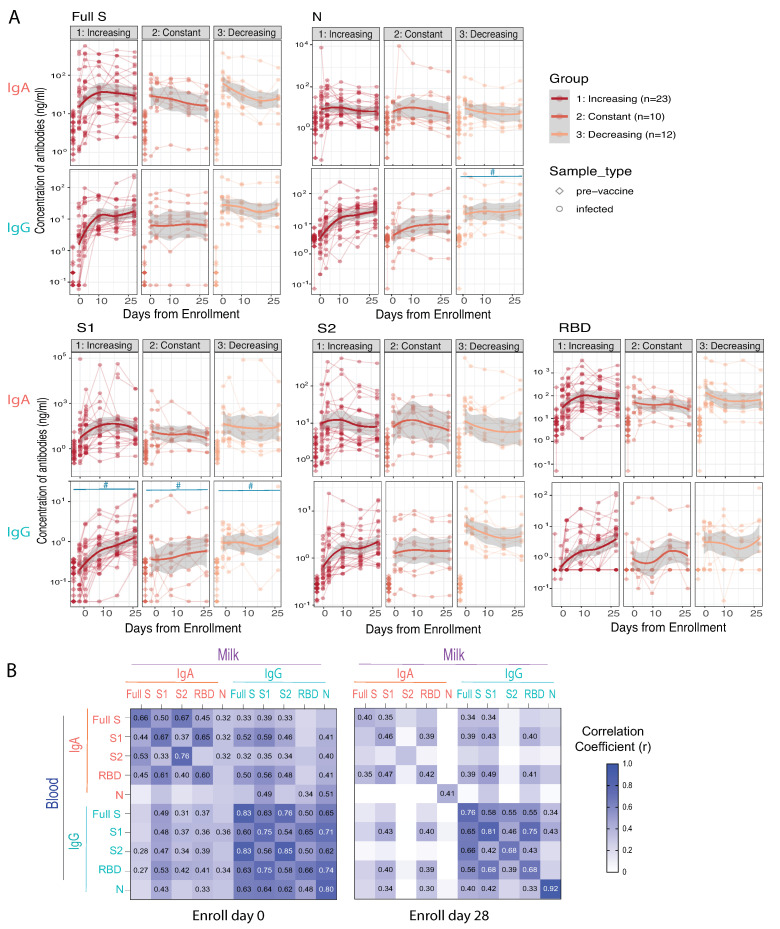
Anti-SARS-CoV-2 IgG and IgA antibody responses in human milk after SARS-CoV-2 infection. The infection cohort was separated into Increasing, Constant or Decreasing groups based on milk IgG antibody level ratios of D28/D0: Increasing with > 1; Constant with 0.8–1.2; Decreasing with ratios < 0.8. (**A**) The milk SARS-CoV-2 specific IgA and IgG antibody levels against full S and N proteins, and S1, S2 and RBD subunits, pre-vaccination and after infection. The antibody levels at all time points post-infection in all groups are significantly higher than that of pre-vaccination controls (*p* < 0.0001) and against all antigens and all group. (**B**) Heatmap of Spearman correlation of milk IgA and IgG antibody levels with blood IgA and IgG antibody levels. Correlation coefficients are color coded as shown in the figure and numerical values are shown when correlation *p*-values < 0.05.

**Figure 4 vaccines-10-00980-f004:**
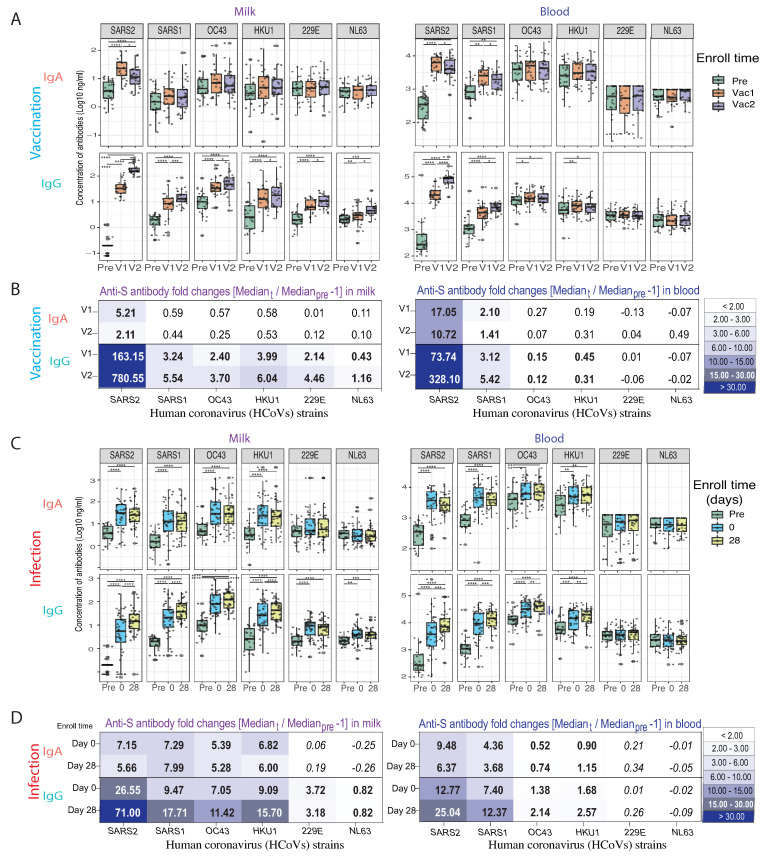
Anti-S IgA and IgG antibody response against human common cold coronaviruses (HCoVs) after SARS-CoV-2 vaccination and infection. The milk and blood HCoV-reactive IgA and IgG antibody levels against spike proteins of SARS-Cov-2 (SARS2), SARS-CoV-1 (SARS1), and HCoVs were measured by multiplex assay. All samples were tested in one batch in duplicate, and significant increases of decreases over time were identified with generalized linear mixed-effects models (**** *p* < 0.0001, *** *p* < 0.001, ** *p* < 0.01, * *p* < 0.05). (**A**) Milk and blood IgA and IgG antibodies against spike proteins of SARS2, SARS1, and HCoVs elicited by COVID-19 vaccination. (**B**) The antibody fold-changes of milk and blood anti-S IgA and IgG antibodies elicited by acute COVID-19 vaccination. (Fold change = [Median post-vac]/[Median pre-vaccination] − 1). Statistically significant changes are indicated in bold (*p* < 0.05). (**C**) Milk and blood anti-S IgA and IgG antibodies elicited by acute COVID-19 infection. (**D**) The antibody fold-changes of milk and blood anti-S IgA and IgG antibodies elicited by acute COVID-19 vaccination. (Fold change = [Median post-infection]/[Median pre-vaccination] − 1). Statistically significant changes are indicated in bold (*p* < 0.05).

**Figure 5 vaccines-10-00980-f005:**
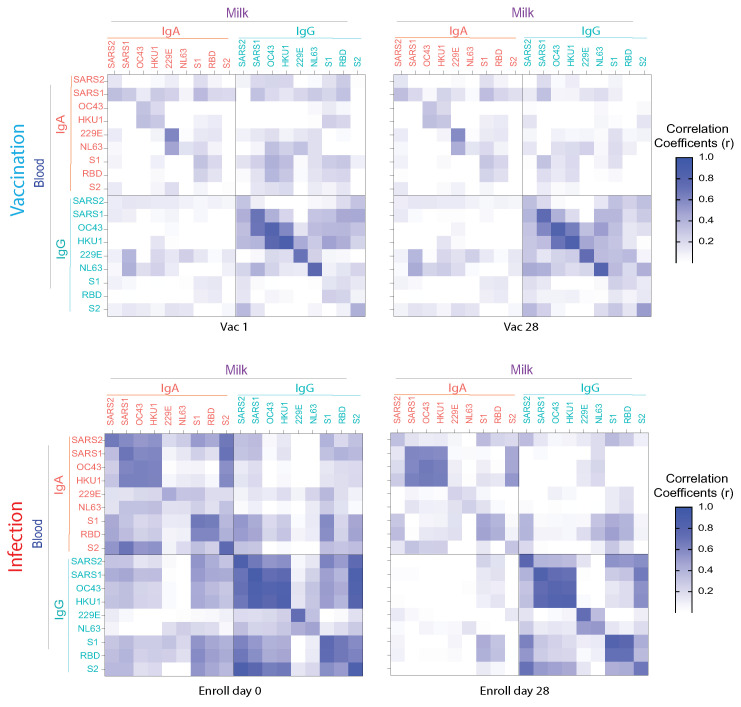
Heatmap of the correlation matrix of antibody responses between milk and blood IgA and IgG against SARS-CoV-2 spike protein (Full-S), S1, S2 and RBD domains, SARS-CoV-1 (SARS1), and HCoVs (OC43, HKU1, 229E, NL63) elicited by vaccination (Vac1 and Vac2) and those that were elicited by acute COVID-19 infection (Day 0 and Day 28). Spearman correlation coefficients (r) between milk and blood immunoglobulins are color coded as indicated in the Figure.

## Data Availability

Data can be accessed from https://figshare.com/s/c0a418db25ed56e163cf, and DOI: 10.6084/m9.figshare.20098235. The access date is 18 June 2022.

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
