# Peer review of "Broad Cross-Reactive IgA and IgG against Human Coronaviruses in Milk Induced by COVID-19 Vaccination and Infection"

_vaccines, 2022, doi:10.3390/vaccines10060980_

Round 1
Reviewer 1 Report
The current pandemic, caused by SARS-CoV-2 infection, has increased the interest in human coronaviruses (CoVs). A deep knowledge of the immune response against human CoVs, including cross-reactive responses, would help to improve public health approaches to deal with these viruses. In that line, the authors build on previous work on the humoral response against SARS-CoV-2 and other human CoVs in both vaccinated and SARS-CoV-2 infected individuals. This manuscript is focused in the antibody levels in human milk and how they correlate with those in blood. The work is well written and, although the novelty of the conclusions is limited, the data would be valuable to increase the scientific knowledge on the humoral immune response against human CoVs.
Specific comments:
1. A point for discussion: since the samples were collected between July 2020 and April 2021, how can VOCs affect the results? Have the authors considered to include different VOCs RBD (or S) in their multiplex assays?
2. Why MERS-CoV was not included in the analysis, as it was in the previous work (reference #11)? Please note that it is also a human betacoronavirus, still circulating, so the data would be valuable.
3. Section 3.4 and lines 235-236. Please note that, after vaccination, IgG levels for SARS-CoV in blood were similar to those in milk, while the levels for the rest of human CoVs were significantly lower than those in milk. The result seems surprising, considering that vaccination follows the intramuscular route, therefore not fully stimulating the mucosal immunity. Is there any explanation for the observed effect? This issue would, at least, be discussed.
4. Fig.4. How do the results change if only neutralizing IgAs and IgGs are considered? Please note that this is relevant to propose that COVID-19 vaccination and infection may provide passive immunity through the milk against common cold CoVs.
In line with this comment, is there any data on whether seasonal common cold CoVs infections have decreased during SARS-CoV-2 pandemic?
Minor comments:
1. Lines 24-26. Please, note that, to follow the taxonomy rules and in agreement with the ICTV, “coronaviridae” should be “Coronaviridae” in italics. In addition, please note that the correct naming and acronyms for the cited CoVs are: SARS-CoV, Middle East respiratory syndrome coronavirus (MERS-CoV), HCoV-OC43, HCoV-HKU1, HCoV-229E and HCoV-NL63
2. Lines 27-28. Please note that the S protein contains the immunodominant epitopes in all CoVs, not just in SARS-CoV-2
3. Line 329. “HCOV” should be “HCoV”
Author Response
Specific comments:
- A point for discussion: since the samples were collected between July 2020 and April 2021, how can VOCs affect the results? Have the authors considered to include different VOCs RBD (or S) in their multiplex assays?
The Variants of Concern (VOCs) of SARS-CoV-2 that appeared during the time of sample collection (July 2020 and April 2021) were B.1.1.7 (Alpha, V1, 20I), B.1.351(Belta, V2, 20H ) and P.1(Gamma, V3, 20J). Based on the GISAID database (https://www.gisaid.org/phylodynamics/global/nextstrain/), The Alpha variant increased from 0 to 37 % by April 13th, 2021, among all isolated virus strains in the North-America. Only 3% Gamma 20J and less than 0.01% Belta 20H had been found by April 2021.
We don’t expect the VOCs will affect the results since studies had shown no difference or a mild decrease in neutralization activity using immune sera from the mRNA-based Pfizer/BioNTech) vaccinated subjects against Alpha 20I viruses or pseudovirus (https://www.frontiersin.org/articles/10.3389/fcimb.2021.781429/full).
After we finished this study, but with more VOCs appearing, we developed the multiplex assay aside from the above VOCs, which also includes the whole S or RBD domain of B.1.617.2 (Delta), B.1.1.529 (Omicron), which had been used for our current COVID-19 vaccination antibody response studies. We are preparing the research report now.
- Why MERS-CoV was not included in the analysis, as it was in the previous work (reference #11)? Please note that it is also a human betacoronavirus, still circulating, so the data would be valuable.
Thanks for mentioning the MERS-CoV and its antibodies. We tested the anti-MERS-CoV S and N antibodies as we did in our previous study. However, the results were similar to our previous publication (reference #11), the MERS-CoV antibodies were always very low, which is not surprising because it was not circulating in the US population. Worldwide, there have only been 2,602 cases of MERS since 2012, and only 2 since 2020, both in Qatar (https://www.ecdc.europa.eu/en/publications-data/geographical-distribution-confirmed-mers-cov-cases-reporting-country-april-2012-1). Thus, MERS data did not provide any significant results or useful information for this study. In order to keep the results clear and easier to follow, we did not report the MERS-CoVs results.
- Section 3.4 and lines 235-236. Please note that, after vaccination, IgG levels for SARS-CoV in blood were similar to those in milk, while the levels for the rest of human CoVs were significantly lower than those in milk. The result seems surprising, considering that vaccination follows the intramuscular route, therefore not fully stimulating the mucosal immunity. Is there any explanation for the observed effect? This issue would, at least, be discussed.
In Section 3.4 and lines 235-236, we mentioned how many times the anti-HCoVs antibody level increased after vaccination. We did not directly compare the IgG levels in the blood with that in the milk, since they are in different log ranges. The IgG anti-SARS-CoV2-S antibody is 104 to 106ng/ml level after vaccination, but the milk only 100 to 102 ng/ml level. Please see Figure 2.
- Fig.4. How do the results change if only neutralizing IgAs and IgGs are considered? Please note that this is relevant to propose that COVID-19 vaccination and infection may provide passive immunity through the milk against common cold CoVs. In line with this comment, is there any data on whether seasonal common cold CoVs infections have decreased during SARS-CoV-2 pandemic?
Thanks for pointing it out, that is a good question. The mPlex assay is the antibody binding assay, and does not directly measure neutralization by IgAs and IgGs. There are some technical difficulties to test the milk neutralizing antibodies, and neutralizing antibodies against HCoVs. However, our previous study (reference #20) (https://jamanetwork.com/journals/jamapediatrics/fullarticle/2786219) showed that the vaccination significantly increases the milk SARS-CoV-2 neutralizing IgG and IgA activities.
Minor comments:
- Lines 24-26. Please, note that, to follow the taxonomy rules and in agreement with the ICTV, “coronaviridae” should be “Coronaviridae” in italics. In addition, please note that the correct naming and acronyms for the cited CoVs are: SARS-CoV, Middle East respiratory syndrome coronavirus (MERS-CoV), HCoV-OC43, HCoV-HKU1, HCoV-229E and HCoV-NL63
Thanks for the comments. Line 24-26 had been changed as the reviewer suggested.
- Lines 27-28. Please note that the S protein contains the immunodominant epitopes in all CoVs, not just in SARS-CoV-2
It had been revised.
- Line 329. “HCOV” should be “HCoV”
Thanks, it had been revised

Reviewer 2 Report
the paper I reviewed is very interesting, focusing on an item that could be of great interest in this pandemic context. data about antibody titers in human milk are new and, to the best of my knowledge, to date they are limited.
in my opinion the paper is well written, metodologically appropriate and shows new and interesting results.
I do not find any particular concern about it, I would only ask to clarify a specific point: in the results section (page 4, line 151) authors say that IgG levels in milk and blood increase up to 187 days post-vaccination but it is not clear if they followed up the enrolled patients up to that period of time or if this date came from a previous published work. It seems that they performed a follow up of their patients up to this time point: I think it would be better to report this information (along with the duration of the follow-up) or to clarify if this information derives from previous or published observations.
Author Response
The response to the reviewer 2:
the paper I reviewed is very interesting, focusing on an item that could be of great interest in this pandemic context. data about antibody titers in human milk are new and, to the best of my knowledge, to date they are limited.
in my opinion the paper is well written, metodologically appropriate and shows new and interesting results.
Thanks for the comments.
I do not find any particular concern about it, I would only ask to clarify a specific point: in the results section (page 4, line 151) authors say that IgG levels in milk and blood increase up to 187 days post-vaccination but it is not clear if they followed up the enrolled patients up to that period of time or if this date came from a previous published work. It seems that they performed a follow up of their patients up to this time point: I think it would be better to report this information (along with the duration of the follow-up) or to clarify if this information derives from previous or published observations.
Thanks for pointing it out. We have clarified this comment. Elevated IgG levels were found up to 187 days in our previous published study (reference #20). Long-time point samples were not evaluated in this study. We revised the statement here to clear it up.
